# Expression of a $CO_2$-permeable aquaporin enhances mesophyll conductance in the $C_4$ species *Setaria viridis*

**Maria Ermakova[1]\*, Hannah Osborn[1], Michael Groszmann[1], Soumi Bala[1], Andrew Bowerman[1], Samantha McGaughey[1], Caitlin Byrt[1], Hugo Alonso-cantabrana[1], Steve Tyerman[2], Robert T Furbank[1], Robert E Sharwood[1,3]\*, Susanne von Caemmerer[1]**

[1]Australian Research Council Centre of Excellence for Translational Photosynthesis, Division of Plant Science, Research School of Biology, Canberra, Australia; [2]ARC Centre of Excellence in Plant Energy Biology, School of Agriculture Food and Wine, University of Adelaide, Adelaide, Australia; [3]Hawkesbury Institute for the Environment, Western Sydney University, Richmond, Australia

\*For correspondence:
maria.ermakova@anu.edu.au
(ME);
r.sharwood@westernsydney.edu.
au (RES)

**Competing interest:** The authors declare that no competing interests exist.

**Abstract** A fundamental limitation of photosynthetic carbon fixation is the availability of $CO_2$. In $C_4$ plants, primary carboxylation occurs in mesophyll cytosol, and little is known about the role of $CO_2$ diffusion in facilitating $C_4$ photosynthesis. We have examined the expression, localization, and functional role of selected plasma membrane intrinsic aquaporins (PIPs) from *Setaria italica* (foxtail millet) and discovered that SiPIP2;7 is $CO_2$-permeable. When ectopically expressed in mesophyll cells of *Setaria viridis* (green foxtail), SiPIP2;7 was localized to the plasma membrane and caused no marked changes in leaf biochemistry. Gas exchange and $C^{18}O^{16}O$ discrimination measurements revealed that targeted expression of SiPIP2;7 enhanced the conductance to $CO_2$ diffusion from the intercellular airspace to the mesophyll cytosol. Our results demonstrate that mesophyll conductance limits $C_4$ photosynthesis at low $pCO_2$ and that SiPIP2;7 is a functional $CO_2$ permeable aquaporin that can improve $CO_2$ diffusion at the airspace/mesophyll interface and enhance $C_4$ photosynthesis.

## Editor's evaluation

The work generated and analysed transformants of Setaria with a putative $CO_2$-permeable aquaporin and found evidence of improved photosynthesis and makes an contribution to understanding if aquaporins can be used to overcome limitations by mesophyll conductance. The new version is improved, making it clear that the variations are robust, but there are still some intriguing open biological questions.

## Introduction

Diffusion of $CO_2$ across biological membranes is a fundamental aspect to photosynthesis. The significant contribution of aquaporins to increased $CO_2$ diffusion has been demonstrated in $C_3$ plants (*Flexas et al., 2006*; *Hanba et al., 2004*; *Sade et al., 2010*). Aquaporins have key roles in regulating the movement of water and solutes into roots and between tissues, cells and organelles (*Tyerman et al., 2021*). These pore-forming integral membrane proteins can be divided into multiple sub-families depending on their amino acid sequence and subcellular localization. The PIPs (plasma membrane intrinsic proteins) are the only sub family, to date, known to permeate $CO_2$ (*Uehlein et al., 2017*). The PIPs are subdivided into paralog groups PIP1s and PIP2s, based on sequence homology (*Azad et al.,*

*2016*; *Chaumont et al., 2001*; *Groszmann et al., 2016*). Typically, PIP2s show higher water permeability when expressed in heterologous systems (*Chaumont et al., 2000*) and PIP1s seemingly require interaction with a PIP2 to correctly traffic to the plasma membrane (*Berny et al., 2016*; *Zelazny et al., 2007*). In plants, a number of $CO_2$ permeable PIPs have been identified including *Arabidopsis thaliana* AtPIP1;2 (*Heckwolf et al., 2011*) and AtPIP2;1 (*Wang et al., 2016*); *Hordeum vulgare* HvPIP2;1, HvPIP2;2, HvPIP2;3, and HvPIP2;5 (*Mori et al., 2014*); *Nicotiana tabacum* NtPIP1;5 (NtAQP1) (*De Rosa et al., 2020*; *Uehlein et al., 2003*) and *Zea mays* ZmPIP1;5 and ZmPIP1;6 (*Heinen et al., 2014*).

The roles of the $CO_2$ permeable aquaporins have been largely characterized in $C_3$ photosynthetic plants where aquaporins localized in both the plasma membrane and the chloroplast envelope have been shown to facilitate $CO_2$ diffusion from the intercellular airspace to the site of Rubisco in chloroplasts (*Kaldenhoff, 2012*; *Uehlein et al., 2008*). The capacity for $CO_2$ diffusion to the initial sites of carboxylation influences the amount of water loss through transpiration (*Cousins et al., 2020*). Therefore, by providing a more efficient pathway for $CO_2$ diffusion, these membrane pores may contribute to increasing the water-use-efficiency (*Groszmann et al., 2016*). However, little is known about the role of $CO_2$ permeable aquaporins and their influence on $CO_2$ diffusion from substomatal cavities to the first site of carboxylation in $C_4$ photosynthesis. The $C_4$ photosynthetic pathway is a biochemical $CO_2$ pump where the initial conversion of $CO_2$ to bicarbonate ($HCO_3^-$) by carbonic anhydrase (CA) and subsequent fixation of phosphoenolpyruvate (PEP) by PEP carboxylase (PEPC) takes place in the cytosol of mesophyll cells. The pathway requires a close collaboration between mesophyll and bundle sheath cells and this constrains leaf anatomy limiting mesophyll surface area that forms a diffusive interface for $CO_2$ (*Evans and Von Caemmerer, 1996*). Mesophyll conductance is defined as the conductance to $CO_2$ diffusion from the intercellular airspace to the mesophyll cytosol (*Evans and Von Caemmerer, 1996*; *Osborn et al., 2017*; *Ubierna et al., 2017*). Although the rate of $C_4$ photosynthesis is almost saturated at ambient $pCO_2$, current modelling suggests that higher mesophyll conductance can increase assimilation rate and water-use-efficiency at low intercellular $CO_2$ partial pressures which occur when stomatal conductance is low (*von Caemmerer and Furbank, 2016*).

*Setaria italica* (foxtail millet) and *Setaria viridis* (green foxtail) are $C_4$ grasses of the Paniceae tribe and Poaceae family, related to important agronomical crops such as *Z. mays* (maize) and *Sorghum bicolor* (sorghum). *S. viridis* is frequently used as a model species for $C_4$ photosynthesis research as it is diploid with a relatively small genome that is sequenced and can be easily transformed (*Brutnell et al., 2010*; *Ermakova et al., 2019*; *Osborn et al., 2017*). Here, we used a yeast heterologous expression system to examine the permeability to $CO_2$ of selected PIPs from *S. italica*. We identified *SiPIP2;7* as encoding a $CO_2$-permeable aquaporin that, when expressed in the plasma membrane of *S. viridis* mesophyll cells, increased mesophyll conductance. Our results demonstrate that $CO_2$-permeable aquaporins can be used to increase $CO_2$ diffusion from the intercellular airspace to mesophyll cytosol to provide higher carboxylation efficiency in $C_4$ leaves.

## Results

### *S. italica* PIP family

Four *PIP1* and eight *PIP2* genes were identified in both *S. italica* and *S. viridis* and their protein sequences were 99–100% identical between the two species (*Supplementary file 1*). Phylogenetic analysis based on the amino acid sequences of the *S. italica* PIP family showed that three distinct clades emerge: the PIP1 clade, PIP2 clade I, and PIP2 clade II (*Figure 1—figure supplement 1*). Isoforms within these three clades have characteristic differences including sequence signatures associated with substrate selectivity (*Supplementary file 2*). Three of SiPIP1s (1;1, 1;2, and 1;5) and all SiPIP2 clade I members (2;1, 2;4, 2;5, 2;6, and 2;7) matched the current consensus sequence for $CO_2$ transport (*Azad et al., 2016*; *Perez Di Giorgio et al., 2014*).

RNA-seq data from the publicly available Phytomine database (Phytozome) was examined for tissue-specific expression patterns of the *S. italica PIPs* (*Figure 1a*). *SiPIP1;1, 1;2, 1;5*, and *2;1* were expressed at moderate to high levels and *SiPIP2;6* at low to moderate levels, in all tissues analyzed (root, leaves, shoot, and panicle). *SiPIP1;6, 2;4, 2;5, 2;7*, and *2;3* were expressed predominantly in roots at low to moderate levels. *SiPIP2;8* was expressed only in leaves and *SiPIP2;2* transcripts were not detected.

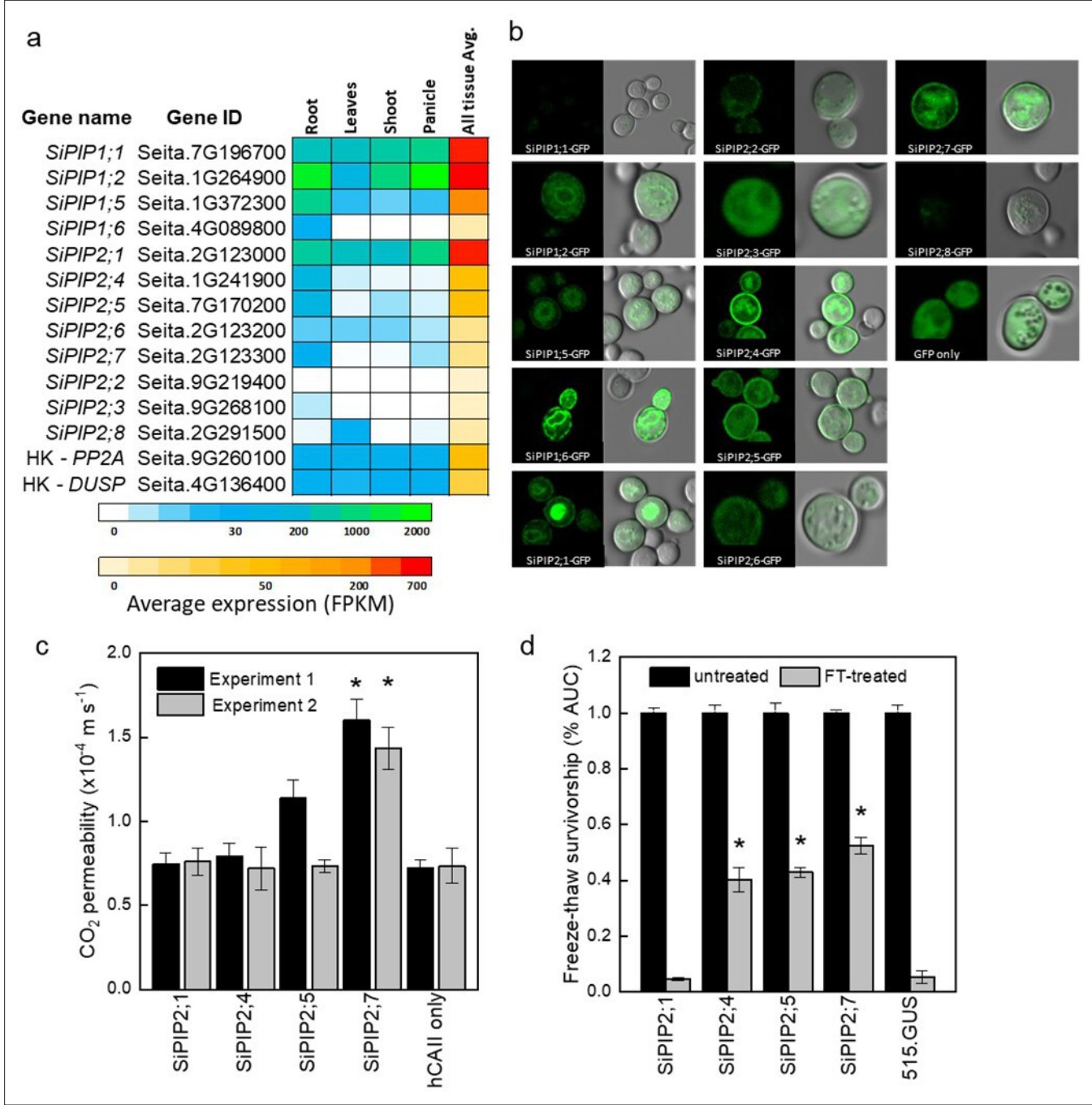

**Figure 1.** Identification of the $CO_2$-permeable aquaporin SiPIP2;7 from *Setaria italica*.
(a) Expression atlas of the *SiPIP* genes generated from Phytomine reported as Fragments Per Kilobase of transcript per Million mapped reads (FPKM). House-keeping genes (HK) *PROTEIN PHOSPHATASE 2A* (*PP2A*) and *DUAL SPECIFICITY PROTEIN* (*DUSP*) were included for reference. (b) Localization of SiPIP-GFP fusions expressed in yeast visualised with confocal microscopy; left panels – GFP fluorescence; right panels – bright field overlaid with GFP fluorescence. Measured cell diameters are shown in *Figure 1—figure supplement 2*. (c) $CO_2$ permeability assay on yeast co-expressing *SiPIPs* and *human CARBONIC ANHYDRASE II* (*hCAII*) analyzed by stopped-flow spectrometry (see *Figure 1—figure supplement 2* for details). 'hCAII only' expression was used as negative control. Mean±SE, *n*=3 biological replicates. Two independent experiments are presented. Asterisks indicate statistically significant differences between yeast expressing *SiPIPs* and 'hCAII only' control (*t*-test, *p*<0.05). (d) Yeast water permeability was assessed in the yeast aquaporin deletion background (*aqy1/2*) by the cumulative growth between untreated and freeze-thawed cells and determined by the percent area under the curve (% AUC). The yeast expressing the β-glucuronidase reporter gene (515.GUS) was used as negative control. Mean±SE, *n*=4 biological replicates. Asterisks indicate statistically significant differences between yeast expressing *SiPIPs* and 515.GUS control (*t*-test, *p*<0.01).

The online version of this article includes the following figure supplement(s) for figure 1:

*Figure 1 continued on next page*

*Figure 1 continued*

**Source data 1.** Gene expression and yeast assays.

**Figure supplement 1.** Phylogenetic analysis of *Setaria italica* PIP protein sequences.

**Figure supplement 2.** Expression of *Setaria italica* PIPs in yeast for functional testing of $CO_2$ permeability.

**Figure supplement 3.** $CO_2$ transport capacity of SiPIP2;7 is not confounded by permeability of protons.

**Figure supplement 4.** SiPIP2;7 is a functional water channel when expressed in *Xenopus laevis* oocytes.

## Functional characterization of SiPIPs

GFP localization of SiPIP-GFP fusions was used to confirm expression and determine targeting to the yeast plasma membrane (*Figure 1b*). Overall, SiPIP1s had lower GFP signal that was patchy at the cell periphery with strong internal signal consistent with localization to the endoplasmic reticulum. GFP signal was also present diffusively throughout the cytosol suggestive of protein degradation. Overall, SiPIP1s were poorly produced in yeast and were not efficiently targeting the plasma membrane as needed for the functional assays. For the PIP2s, only SiPIP2;1, SiPIP2;4, SiPIP2;5, and SiPIP2;7 showed clear localization to the plasma membrane in addition to other internal structures, and were therefore selected for further functional analyses.

$CO_2$ permeability was measured in yeast co-expressing a *SiPIP* along with *human CARBONIC ANHYDRASE II* (*hCAII*). A stopped-flow spectrophotometer was used to monitor $CO_2$-triggered intracellular acidification via changes in fluorescence intensity of a pH-sensitive fluorescein dye (*Figure 1—figure supplement 2*; *Ding et al., 2013*; *Heckwolf et al., 2011*; *Uehlein et al., 2008*). Importantly for reliable results, all SiPIP yeast lines tested showed similar cell volumes and were not limited by CA activity (*Figure 1—figure supplement 2*). A screen of the lines revealed that yeast expressing *SiPIP2;7* had the highest $CO_2$ permeability of $1.5 \times 10^{-4}$ m s$^{-1}$, which was significantly larger than the negative control expressing *hCAII* only (*Figure 1c*). Other *SiPIP*s displayed comparable $CO_2$ permeability to the *hCAII* only control. The changes in $CO_2$ permeability detected on the stopped-flow spectrophotometer for yeast expressing *SiPIP2;7* were not an artifact brought on by an increased permeability to protons causing the intracellular acidification (*Figure 1—figure supplement 3*).

Freeze-thaw survival assays, which quantify water permeability of aquaporins (*Tanghe et al., 2002*), provided further confirmation that the SiPIPs expressed in yeast were functional. Overexpression of water permeable aquaporins greatly improves freeze-thaw tolerance in yeast, especially in the highly compromised aquaporin knockout mutant *aqy1/2* (*Tanghe et al., 2002*). Yeast expressing the β-glucuronidase reporter gene (515.GUS) was used a control to show that the single freeze-thaw treatment was effective in almost killing off the entire yeast population (*Figure 1d*). Consistent with the poor plasma membrane localization and abundance of SiPIP2;1-GFP (*Figure 1b*), yeast expressing *SiPIP2;1* did not show any protection to freeze-thaw treatments (*Figure 1c*). On the other hand, *SiPIP2;4, 2;5*, and *2;7* all showed some level of protection, indicating that they permeated water and were functional within the plasma membrane of yeast cells. For detailed characterization of water permeability, SiPIP2;7 was expressed in *Xenopus laevis* oocytes. Swelling assay confirmed that SiPIP2;7 is a functional water channel (*Figure 1—figure supplement 4*).

## Expression of SiPIP2;7 in mesophyll cells of *S. viridis*

To confirm and exploit the $CO_2$ permeability characteristic of SiPIP2;7 in planta, we created transgenic *S. viridis* plants expressing *SiPIP2;7* with a C-terminal FLAG-tag fusion and under the control of the mesophyll-preferential *Z. mays* PEPC promoter (*Gupta et al., 2020*; *Salesse-Smith et al., 2018*). Out of 52 $T_0$ plants analyzed for SiPIP2;7-FLAG protein abundance and the hygromycin phosphotransferase (*hpt*) gene copy number (*Figure 2—figure supplement 1*), lines 27, 44, and 52 were selected for further analysis because they had the strongest FLAG signal per transgene insertion number. Immunodetection of FLAG and photosynthetic proteins was performed on leaves of homozygous transgenic plants (*Figure 2a*); azygous plants of line 44 were used as control hereafter. Monomeric and dimeric SiPIP2;7-FLAG was detected in all transgenic plants (*Figure 2—figure supplement 1*) and abundance of the prevalent dimeric form was used for relative quantification of SiPIP2;7 abundance (*Figure 2a*). Plants of line 44 had the highest production of SiPIP2;7-FLAG whilst plants of

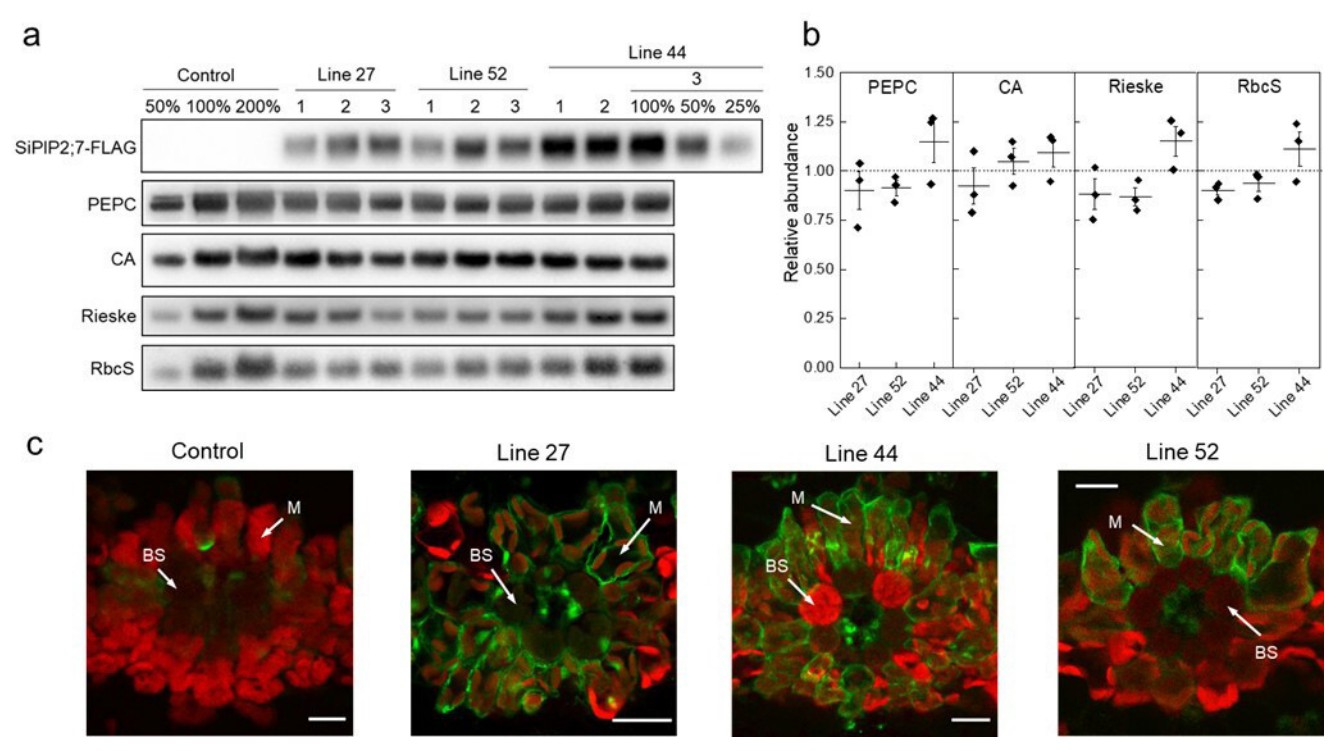

**Figure 2.** Characterization of *Setaria viridis* plants expressing *SiPIP2;7-FLAG* in mesophyll cells.
 (**a**) Immunodetection of SiPIP2;7-FLAG and photosynthetic proteins in leaf protein samples loaded on leaf area basis. Three plants from each of the three transgenic lines were analyzed and dilution series of the control and line 44-3 samples were used for relative quantification. (**b**) Protein abundances calculated from the immunoblots relative to control plants. Mean±SE. No significant difference was found between the transgenic and control plants (*t*-test). (**c**) Immunolocalization of SiPIP2;7-FLAG on leaf cross-sections visualized with confocal microscopy. Fluorescence signals are pseudo-colored: green – FLAG antibodies labelled with secondary antibodies conjugated with Alexa Fluor 488; red – chlorophyll autofluorescence. BS, bundle sheath cell; M, mesophyll cell. Scale bars = 20 µm. Azygous plants of line 44 were used as control. Uncropped images of the blots are provided in *Figure 2— source data 1*.

The online version of this article includes the following figure supplement(s) for figure 2:

**Source data 1.** Uncropped images of western blots.

**Figure supplement 1.** Immunodetection of SiPIP2;7-FLAG in leaves of transgenic *Setaria viridis* plants.

**Figure supplement 1—source data 1.** Uncropped images of western blots.

**Figure supplement 2.** qPCR analysis of *SiPIP2;6* and *SiPIP2;7* expression in leaves and roots of control and transgenic *Setaria viridis* plants.

lines 27 and 52 accumulated about 2–4 times less of this protein. Immunodetection of FLAG on leaf cross-sections, visualized with confocal microscopy, confirmed partial localization of SiPIP2;7-FLAG to the plasma membrane of mesophyll cells (*Figure 2c*). Transcript analysis confirmed highly elevated expression of *SiPIP2;7-FLAG* in leaves but not in roots of transgenic lines and showed no changes in expression level of closely related *SiPIP2;6* (*Figure 2—figure supplement 2*).

Abundances of photosynthetic proteins PEPC, CA, the Rieske subunit of the Cytochrome $b_6f$ complex, and the small subunit of Rubisco (RbcS), did not differ between transgenic and control plants (*Figure 2a*). In line with the immunoblotting results, measured activities of PEPC and CA, and the amount of Rubisco active sites were not altered in the transgenic plants (*Table 1*). Chlorophyll content, leaf dry weight per area, and biomass of roots and shoots did not differ between the geno-types either (*Table 1*).

To study the impact of *SiPIP2;7-FLAG* expression on the photosynthetic properties in transgenic plants, we conducted concurrent gas exchange and chlorophyll fluorescence analyses at different intercellular $CO_2$ partial pressures ($C_i$) (*Figure 3*). No significant changes were detected between transgenic and control plants in $CO_2$ assimilation rates (*A*), effective quantum yield of Photosystem II

**Table 1.** Properties of *Setaria viridis* plants expressing *SiPIP2;7-FLAG* in mesophyll cells.
PEPC, PEP carboxylase; Rubisco, ribulose bisphosphate carboxylase oxygenase; LMA, leaf mass per area. Azygous plants of line 44 were used as control. Mean±SE, *n*=3 except for biomass (*n*=8). Three-weeks old plants before flowering were used for all analyses. No significant difference was found between the transgenic and control plants (one-way ANOVA, $\alpha$=0.05).

| Parameter | Control | Line 27 | Line 44 | Line 52 |
|---|---|---|---|---|
| PEPC activity, µmol $CO_2$ m$^{-2}$ s$^{-1}$ | 220.1±25.8 | 197.6±12.7 | 208.7±7.9 | 218.5±3.5 |
| CA hydration rate, mol m$^{-2}$ s$^{-1}$ bar$^{-1}$ | 6.50±0.10 | 6.32±0.22 | 5.34±0.67 | 5.35±0.56 |
| Rubisco active sites, µmol m$^{-2}$ | 12.17±0.63 | 12.53±0.54 | 12.84±0.13 | 12.63±0.74 |
| Chlorophyll (*a*+*b*), mmol m$^{-2}$ | 0.71±0.07 | 0.72±0.04 | 0.72±0.05 | 0.72±0.08 |
| Chlorophyll *a/b* | 5.01±0.16 | 5.08±0.05 | 4.97±0.09 | 5.07±0.15 |
| LMA, g (dry weight) m$^{-2}$ | 23.6±1.6 | 24.0±1.5 | 25.6±1.3 | 25.4±1.3 |
| Shoot biomass, g (dry weight) plant$^{-1}$ | 2.06±0.36 | 2.01±0.20 | 2.23±0.31 | 2.24±0.34 |
| Root biomass, g (dry weight) plant$^{-1}$ | 0.27±0.07 | 0.28±0.03 | 0.34±0.06 | 0.35±0.05 |

($\phi$ PSII) or stomatal conductance to water vapor (*Figure 3—figure supplement 1*). The SiPIP2;7-FLAG protein abundance was compared to the gas exchange phenotype in individual plants (*Figure 3—figure supplement 2*). A statistically significant polynomial relationship ($R^2$=0.345, *p*<0.05) was found between the initial slopes and the relative protein content, which was significantly better than that achieved using a linear model (*p*<0.05). No significant relationship was observed between the SiPIP2;7-FLAG abundance and the saturating rates of assimilation ($A_{max}$; *Figure 3—figure supplement 2*).

## Mesophyll conductance to $CO_2$ in plants expressing SiPIP2;7

Next, we analyzed in detail the initial slopes of the $AC_i$ curves and mesophyll conductance. Fitting linear regressions indicated that mean±SE values of the initial slopes of the $AC_i$ curves for lines 27, 44, and 52 were 0.46±0.03, 0.52±0.01, and 0.53±0.05, respectively, compared to the value of 0.41±0.02 in control plants (*Figure 4a*). Measurements of $\Delta^{18}O$ were used to estimate conductance of $CO_2$ from the intercellular airspace to the sites of $CO_2$ and $H_2O$ exchange in the mesophyll cytosol ($g_m$) with the assumption that $CO_2$ was in full isotopic equilibrium with leaf water in the cytosol (*Barbour et al., 2016*; *Osborn et al., 2017*). Transgenic lines showed mesophyll conductance of 0.59±0.05, 0.55±0.08, and 0.46±0.04 mol m$^{-2}$ s$^{-1}$ bar$^{-1}$ compared to the mean±SE value of 0.42±0.03 mol m$^{-2}$ s$^{-1}$ bar$^{-1}$ in control plants (*Figure 4b*). Two-way ANOVA analysis with Tukey post hoc test on the initial slopes of the $AC_i$ curves and $g_m$ showed that differences measured in plants of lines 27 and 44 were statistically significant from the control plants (*p*=0.04573 and 0.03724, respectively). Interestingly, for plants of line 52, only initial slopes were significantly different compared to control plants when compared by one-way ANOVA (Tukey post hoc test, $\alpha$=0.05, *p*=0.02993).

We also used the $g_m$ calculations proposed by *Ogée et al., 2018* which try to account for the rates of bicarbonate consumption by CA. The CA hydration constant ($k_{CA}$) of 6.5 mol m$^{-2}$ s$^{-1}$ bar$^{-1}$ was used for these calculations (*Table 1*). We found that the $g_m$ measured with this method gave on average 1.25 times greater values but did not change the ranking of mesophyll conductance shown in *Figure 4a* (*Figure 4—figure supplement 1*). The $C_4$ photosynthetic model by *von Caemmerer and Furbank, 1999* and *von Caemmerer, 2000* relates the initial slope of the $CO_2$ response curve (d$A$/$C_i$) to $g_m$ (see *Figure 4* caption and Materials and methods). *Figure 4c* shows that the measured relationship between the initial slope and $g_m$ fits closely with model prediction.

## Discussion

The diffusion of $CO_2$ from the Earth's atmosphere to the site of primary carboxylation within leaves of $C_3$ and $C_4$ plants often limits photosynthesis and impacts the efficient use of water. Overexpression of $CO_2$ permeable aquaporins with the aim to enhance leaf $CO_2$ diffusion has been extensively probed in $C_3$ plants. Many studies have demonstrated that it was an effective strategy to improve $g_m$, leading to increased assimilation rate or grain yield (*Flexas et al., 2006*; *Hanba et al., 2004*; *Uehlein et al., 2003*;

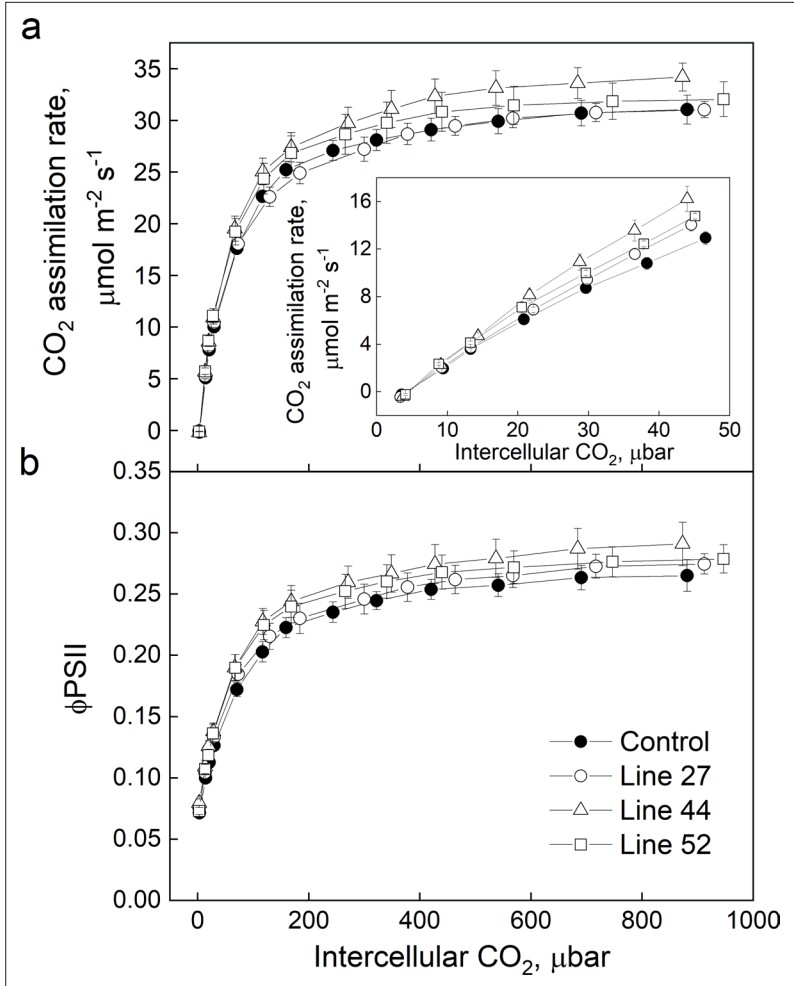

**Figure 3.** $CO_2$ response of $CO_2$ assimilation rate (**a**) and quantum yield of Photosystem II (**b**) in *Setaria viridis* plants expressing *SiPIP2;7-FLAG* in mesophyll cells.
 Measurements were performed at the irradiance of 1500 µmol m$^{-2}$ s$^{-1}$; azygous plants of line 44 were used as control. Mean±SE, *n*=4–6 biological replicates. No significant difference was found between the transgenic and control plants (one-way ANOVA, α=0.05).

The online version of this article includes the following figure supplement(s) for figure 3:

**Source data 1.** Gas exchange and fluorescence analysis.

**Figure supplement 1.** Stomatal conductance to water vapor (gsw) measured at different intercellular $CO_2$ partial pressure.

**Figure supplement 2.** The relationship between the relative abundance of SiPIP2;7 and the initial slopes or the maximum assimilation rates ($A_{max}$) of the $AC_i$ curves in individual $T_1$ plants of four transgenic lines.

**Figure supplement 2—source data 1.** Uncropped images of western blots.

---

*Xu et al., 2019*). However, some published research could not identify a functional link between the $CO_2$ permeable aquaporins and mesophyll conductance, and possible explanations include functional redundancy between aquaporin isoforms (*Kromdijk et al., 2020*) and complications of expressing membrane proteins in the chloroplast envelope (*Fernández-San Millán et al., 2018*). In contrast to C$_3$ plants, where $CO_2$ needs to cross both plasma membrane and the chloroplast envelope to reach the site of carboxylation, the primary carboxylation step in C$_4$ plants occurs in the cytosol of mesophyll cells. Due to high photosynthetic rates of C$_4$ plants, this generates a large $CO_2$ drawdown between the intercellular airspaces and the cytosol and thus a large mesophyll conductance (*Evans and Von Caemmerer, 1996*). Because only one membrane needs to be traversed by $CO_2$, C$_4$ plants could serve as a simpler model to demonstrate the effect of $CO_2$ permeable aquaporins on photosynthesis.

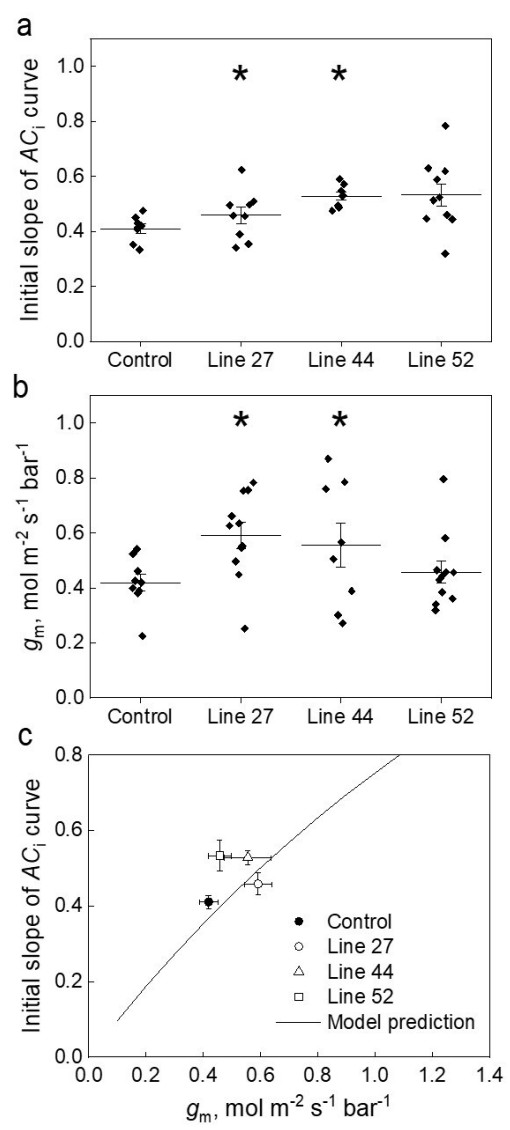

**Figure 4.** Effect of the mesophyll conductance, $g_m$, on the initial slope of the $CO_2$ assimilation response curve to the intercellular $CO_2$ partial pressure ($AC_i$ curve) in leaves of *Setaria viridis* expressing *SiPIP2;7-FLAG* in mesophyll cells. (**a**) Initial slope of the $AC_i$ curves estimated by linear fitting of curves (a subset of curves is presented in **Figure 3a** inset). (**b**) Mesophyll conductance, $g_m$, estimated by oxygen isotope discrimination assuming full isotopic equilibrium (**Osborn et al., 2017**). Measurements were made at ambient $CO_2$ and low $O_2$. (**c**) Data from (**a**) and (**b**) compared to the $C_4$ biochemical model predictions (**von Caemmerer, 2000**; **von Caemmerer and Furbank, 1999**). All graphs show Mean±SE; azygous plants of line 44 were used as control. The model relates the initial slope of the $AC_i$ curve ($dA/C_i$) to $g_m$ by: $\frac{dA}{dC_i} = g_m V_{pmax} / \left( g_m K_p + V_{pmax} \right)$, where $V_{pmax}$ and $K_p$ denote the maximum PEPC activity and the Michaelis Menten constant for $CO_2$ taken here as 250 µmol m⁻² s⁻¹ and 82 µbar (**DiMario and Cousins, 2019**;

*Figure 4 continued*

**von Caemmerer, 2021**). Asterisks indicate statistically significant differences between the plants of lines 27 ($p=0.04573$) and 44 ($p=0.03724$) and control plants (two-way ANOVA with Tukey post hoc test, α=0.05). Plants of line 52 were not significantly different from the control plants ($p=0.27518$).

The online version of this article includes the following figure supplement(s) for figure 4:

**Figure supplement 1.** Comparison of the $g_m$ estimated by oxygen isotope discrimination assuming full isotopic equilibrium (**Osborn et al., 2017**) and calculations suggested by **Ogée et al., 2018**.

Screening *S. italica* PIPs for $CO_2$ permeability in a yeast heterologous system resulted in identification of SiPIP2;7 as a $CO_2$ pore (**Figure 1c**). Expression analysis revealed that *SiPIP2;7* was almost exclusively expressed in roots under ideal conditions (**Figure 1a**, **Figure 2—figure supplement 2**) which, combined with the water permeability identified in yeast and oocyte assays (**Figure 1d**, **Figure 1—figure supplement 4**), suggest that SiPIP2;7 may function in regulating root hydraulic conductivity, a role extensively documented for PIP aquaporins (**Gambetta et al., 2017**; **McGaughey et al., 2018**). The physiological relevance of SiPIP2;7's $CO_2$ permeating capacity is not immediately clear. Gas uptake by roots is well documented (**Stemmet et al., 1962**) and in $C_3$ plants $CO_2$ uptake by roots may contribute to the $C_4$ photosynthesis-like metabolism detected in stems and petioles (**Hibberd and Quick, 2002**). It is possible that *SiPIP2;7* is conditionally expressed in leaves, or even that its capacity to transport $CO_2$ is inadvertent and related to the transportation of another yet undetermined substrate; analogous to the uptake of toxic metalloids by some NIP aquaporins due to their capacity to transport boron (**Mukhopadhyay et al., 2014**). Further work is needed to determine whether PIPs in general function natively as relevant $CO_2$ pores in $C_4$ leaves.

We employed the $CO_2$ transport capacity of SiPIP2;7 to enhance transmembrane $CO_2$ diffusion from the intercellular airspace into the mesophyll cytosol, where CA and PEPC reside, by overexpressing *SiPIP2;7* in *S. viridis*. We confirmed the localization of SiPIP2;7 within the mesophyll plasma membranes (**Figure 2c**) and detected the increase in $CO_2$ diffusion across the mesophyll membranes in transgenic plants by two independent methods. First, we calculated $g_m$ from the $C^{18}O^{16}O$ discrimination measurements (**Figure 4b**) and the theory for these calculations

has been outlined (*Barbour et al., 2016*; *Ogée et al., 2018*; *Osborn et al., 2017*). Second, we fitted linear regressions to the initial slopes of the $AC_i$ curves (*Figure 4a*), which depend on $g_m$, $V_{pmax}$, and $K_p$ where the two latter parameters denote the maximum PEPC activity and the Michaelis Menten constant of PEPC for $HCO_3^-$ (*von Caemmerer, 2000*; *von Caemmerer and Furbank, 1999*). Since PEPC and CA activities were not altered in plants expressing *SiPIP2;7* (*Table 1*), higher initial slopes of the $AC_i$ curves in transgenic lines were attributed to the increased $g_m$. When plotted against each other, the initial slopes and $g_m$ in transgenic and control plants, fitted the model predictions confirming the hypothesized functional role of $g_m$ in $C_4$ photosynthesis (*Pfeffer and Peisker, 1995*; *Ubierna et al., 2017*; *von Caemmerer, 2000*). Importantly, and in line with the model predictions, expression of SiPIP2;7 had an effect only on the initial slopes of the $AC_i$ curves but not on the saturating rates of assimilation (*Figure 3—figure supplement 2*).

Our findings demonstrate that $CO_2$ permeable aquaporins can enhance $CO_2$ diffusion at the airspace/mesophyll interface in $C_4$ plants. However, overexpression of aquaporins is likely accompanied by confounding factors which could explain the lack of phenotype observed from the plants with the highest SiPIP2;7 expression level (*Figure 3—figure supplement 2*) and plants of line 52 showing significant increase of only initial slopes (*Figure 4*). These factors possibly include the transport of multiple substrate(s) by SiPIP2;7 and different efficiency of SiPIP2;7 targeting to the plasma membrane between the individual lines, plants, leaves, and even between different mesophyll cells. Moreover, it is still not clear whether there are other $CO_2$-permeable aquaporins present in *S. viridis* leaves that account for the basal $g_m$ level of $0.42 \pm 0.03$ mol m$^{-2}$ s$^{-1}$ bar$^{-1}$ in control plants. Further research is required to unravel the aforementioned confounding factors. Nevertheless, building on our findings that increasing mesophyll conductance is possible with overexpression of SiPIP2;7, this trait will be a strong candidate to combine with complementary traits such as the overexpressing of Cytochrome $b_6 f$ (*Ermakova et al., 2019*) and Rubisco (*Salesse-Smith et al., 2018*), which offer improvements to photosynthesis in $C_4$ plants.

## Materials and methods
### Heterologous expression in yeast

cDNAs encoding the 12 *S. italica* aquaporins (*Supplementary file 1*) and *human CARBONIC ANHYDRASE II* (*hCAII*, AK312978) were codon-optimized for expression in yeast with IDT DNA tool (https://sg.idtdna.com/pages/tools) and a yeast-related Kozak sequence was added at the 5′ end to help increase translation (*Nakagawa et al., 2008*). For $CO_2$ permeability measurements, pSF-TPI1-URA3 with an aquaporin and pSF-TEF1-LEU2 with *hCAII* were co-transformed into the *S. cerevisiae* strain INV*Sc*1 (Thermo Fisher Scientific, Waltham, MA). For water permeability measurements, pSF-TPI1-URA3 with an aquaporin was transformed into the *aqy1/2* double mutant yeast strain deficient in aquaporins (*Suga and Maeshima, 2004*). The yeast vectors pSF-TPI1-URA3 and pSF-TEF1-LEU2 were obtained from Oxford Genetics (Oxford, UK). Yeast transformation was performed using the Frozen-EZ yeast transformation II kit (Zymo Research, Irvine, CA) and selection of positive transformants was based on amino acid complementation. To ensure CA was not limiting, CA activity was determined using a membrane inlet mass spectrometry as described by *Endeward et al., 2006* (*Figure 1—figure supplement 2*). For $CO_2$ permeability measurements, an average cell diameter of 4.63 μm was determined by measuring ~100 yeast cells expressing each aquaporin (*Figure 1—figure supplement 2*). To study the subcellular localizations of aquaporins in yeast, a C-terminus GFP tag was added to the sequences into the pSF-TPI1-URA3 vector (pSF-TPI1-URA3-GFP). The fluorescence signal was observed using a Zeiss 780 confocal laser scanning microscope (Zeiss, Oberkochen, Germany): excitation 488 nm and emission 530 nm. Cytosolic GFP expression was used as control.

### $CO_2$ induced intracellular acidification assay

$CO_2$ intracellular acidification was measured in yeast cells loaded with fluorescein diacetate (Sigma-Aldrich, St. Louis, MO) as described previously (*Bertl and Kaldenhoff, 2007*; *Otto et al., 2010*). Briefly, an overnight culture of yeast cells was collected and resuspended in an equal volume of 50 mM 4-(2-hydroxyethyl)-1-piperazineethanesulfonic acid (HEPES)-NaOH, pH 7.0, 50 μM fluorescein diacetate and incubated for 30 min in the dark at 37°C. The suspension was centrifuged and the pellet resuspended in ice-cold incubation buffer (25 mM HEPES-NaOH, pH 6.0, 75 mM NaCl). Cells loaded

with fluorescein diacetate were then injected into the stopped-flow spectrophotometer (DX.17MV; Applied Photophysics, Leatherhead, UK) alongside a buffer solution (25 mM HEPES, pH 6.0, 75 mM NaHCO$_3$, bubbled with CO$_2$ for 2 hr). The kinetics of acidification was measured at 490 nm excitation and >515 nm emission (OG515 long pass filter, Schott, supplied by Applied Photophysics). Data were collected over a time interval of 0.2 s and analyzed using ProData SX viewer software (Applied Photophysics). CO$_2$ permeability was determined using the method of *Yang et al., 2000*. An average of 75 injections over at least three separate cultures was used for each aquaporin.

## Determination of water permeability

A freeze-thaw yeast assay was used to determine water permeability of aquaporins expressed in *aqy1/2* based on previous reports (*Tanghe et al., 2002*). Briefly, an overnight culture was diluted to ~6×10$^6$ cells (final volume 1 ml) in appropriate selection liquid growth medium and incubated at 30°C for 1 hr. 250 µl of each culture were then aliquoted into two standard 1.5 ml microtubes: the first (control) tube was placed on ice and the second tube was subject to a single freeze-thaw treatment, consisting of 30 s freezing in liquid nitrogen and thawing for 20 min in a 30°C water bath. Following the treatment, the cells were placed on ice. The tubes were then vortexed briefly to ensure even suspension of cells and 200 µl of the culture was transferred to wells of a Nunc-96 400 µl flat bottom untreated plate (Thermo Fisher Scientific, Cat#243656). Yeast growth in control and treated cultures were monitored over a 24–30 hr period in an M1000 Pro plate reader (TECAN, Männedorf, Switzerland) at 30°C with double orbital shaking at 400 rpm and measuring absorbance at 650 nm every 10 min. Growth data were log-transformed and freeze-thaw survival calculated as the growth (area under the curve) of treated culture relative to its untreated control from time zero up until the untreated control culture reached stationary phase.

For swelling assays, the coding sequence of *SiPIP2;7* was cloned into pGEMHE oocyte expression vector using LR clonase II (Thermo Fisher Scientific) and cRNA was synthesized with mMessage mMachine T7 Transcription Kit (Thermo Fisher Scientific). *X. laevis* oocytes were injected with 46 nl of RNAse-free water with either no cRNA or 23 ng cRNA with a micro-injector Nanoinject II (Drummond Scientific, Broomall, PA). Post-injection oocytes were stored at 18°C in a Low Na$^+$ Ringer's solution (62 mM NaCl, 36 mM KCl, 5 mM MgCl$_2$, 0.6 mM CaCl$_2$, 5 mM HEPES, 5% [v/v] horse serum [H-1270, Sigma-Aldrich] and antibiotics: 0.05 mg ml$^{-1}$ tetracycline, 100 units ml$^{-1}$ penicillin/0.1 mg ml$^{-1}$ streptomycin), pH 7.6 for 24–30 hr. Photometric swelling assay was performed 24–30 hr post-injection (*Qiu et al., 2020*).

## Construct assembly and *S. viridis* transformation

The coding sequence of *S. viridis* PIP2;7 (Sevir.2G128300.1, Phytozome, https://phytozome.jgi.doe.gov/) has been codon optimized for the Golden Gate cloning (*Engler et al., 2014*) and translationally fused with the glycine linker and the FLAG-tag coding sequence (*Hopp et al., 1988*). The resulting coding sequence was assembled with the *Z. mays* PEPC promoter and the bacterial tNos terminator into the second expression module of the pAGM4723 binary vector. The first expression module has been occupied by the hygromycin phosphotransferase (*hpt*) gene assembled with the *Oryza sativa Actin-1* promoter and the tNos terminator. The construct was transformed into *S. viridis* cv. MEO V34-1 using *Agrobacterium tumefaciens* strain *AGL1* following the procedure described in *Osborn et al., 2017*. T$_0$ plants resistant to hygromycin were transferred to soil and analyzed for SiPIP2;7-FLAG protein abundance and *hpt* insertion number by droplet digital PCR (iDNA Genetics, Norwich, UK). Lines 27, 44, and 52 were selected for further analysis because they had the strongest FLAG signal per transgene insertion number (*Figure 2—figure supplement 1*). The T$_1$ and T$_2$ progenies of T$_0$ plants 27, 44, and 52 were analyzed. Azygous T$_1$ plants of line 44 and their progeny were used as control.

## Plant growth conditions

Seeds were surface-sterilized and germinated on medium (pH 5.7) containing 2.15 g L$^{-1}$ Murashige and Skoog salts, 10 ml L$^{-1}$ 100× Murashige and Skoog vitamins stock, 30 g L$^{-1}$ sucrose, 7 g L$^{-1}$ Phytoblend, 20 mg L$^{-1}$ hygromycin (no hygromycin for azygous plants). Seedlings that developed secondary roots were transferred to 0.6 L pots with garden soil mix layered on top with 2 cm seed raising mix (Debco, Tyabb, Australia) both containing 1 g L$^{-1}$ Osmocote (Scotts, Bella Vista, Australia). Plants were grown in controlled environmental chambers with 16 hr light/8 hr dark, 28°C day, 22°C night,

60% humidity, and ambient $CO_2$ concentrations. Light intensity of 300 µmol m$^{-2}$ s$^{-1}$ was supplied by 1000 W red sunrise 3200K lamps (Sunmaster Growlamps, Solon, OH). Youngest fully expanded leaves of the 3–4 weeks plants before flowering were used for all analyses.

## Chlorophyll and enzyme activity

Chlorophyll content was measured on frozen leaf discs homogenized with a TissueLyser II (Qiagen, Venlo, The Netherlands) (*Porra et al., 1989*). PEPC activity was determined after *Pengelly et al., 2010* from fresh leaf extracts from the plants adapted for 1 hr to 800 µmol photons m$^{-2}$ s$^{-1}$. CA activity was measured on a membrane inlet mass spectrometer as a rate of $^{18}O$ exchange from labeled $^{13}C^{18}O_2$ to $H_2^{16}O$ at 25°C according to *Von Caemmerer et al., 2004* by calculating the hydration rate after *Jenkins et al., 1989*. The amount of Rubisco active sites was determined by [$^{14}C$] carboxyarabinitol bisphosphate binding as described earlier (*Ruuska et al., 2000*).

## RNA isolation and qPCR

Leaf and root tissue were frozen in liquid $N_2$. Leaf samples were homogenized using a TissueLyser II and RNA was extracted using the RNeasy Plant Mini Kit (Qiagen). Roots were ground with mortar and pestle in liquid $N_2$ and RNA was isolated according to *Massey, 2012*. Briefly, 150 µl of pre-heated (60°C) extraction buffer (0.1 M trisaminomethane (Tris)-HCl, pH 8.5 mM ethylenediaminetetraacetic acid [EDTA], 0.1 M NaCl, 0.5% sodium dodecyl sulfate [SDS], 1% 2-mercaptoethanol) was added to ~100 mg of fine root powder and incubated at 60°C for 5 min. 150 µl of phenol:chloroform:iso-amyl alcohol (25:24:1) saturated with 10 mM Tris (pH 8.0) and 1 mM EDTA was added to the samples, vortexed vigorously for 10 min and centrifuged at 4500×*g* for 15 min. Aqueous phase was mixed with 120 µl of isopropanol and 15 µl of 3 M sodium acetate and incubated at –80°C for 15 min, then centrifuged at 4500×*g* (30 min, 4°C). The pellet was washed two times in 300 µl of ice-cold 70% ethanol, air-dried, and dissolved in 60 µl of RNase-free water. After addition of 40 µl of 8 M LiCl, samples were incubated overnight at 4°C. Nucleic acids were pelleted by centrifugation at 16,000×*g* (60 min, 4°C), washed two times with 200 µl of ice-cold 70% ethanol, air-dried, and dissolved in RNase-free water. DNA from the samples was removed using an Ambion TURBO DNA-free Kit (Thermo Fisher Scientific), and RNA quality was determined using a NanoDrop (Thermo Fisher Scientific). 100 ng of total RNA were reverse transcribed into cDNA using a SuperScript III Reverse Transcriptase (Thermo Fisher Scientific). qPCR and melt curve analysis were performed on a Viia7 Real-Time PCR System (Thermo Fisher Scientific) using the Power SYBR Green PCR Master Mix (Thermo Fisher Scientific) according to the manufacturer's protocol. Primer pairs designed to distinguish between *S. viridis PIP2;6* and *PIP2;7* using Primer3 in Geneious Prime (https://www.geneious.com) and reference primers are listed in *Supplementary file 3*.

## Western blotting and immunolocalization

Protein isolation from leaves and gel electrophoresis were performed as described earlier (*Ermakova et al., 2019*). Proteins were probed with antibodies against FLAG (ab49763, 1:5000, Abcam, Cambridge, UK), RbcS (*Martin-Avila et al., 2020*) (1:10,000), Rieske (AS08 330, 1:3000, Agrisera, Vännäs, Sweden), PEPC (AS09 458, 1:10,000, Agrisera), CA (*Azad et al., 2016*; *Ludwig et al., 1998*) (1:10,000). Quantification of immunoblots was performed with Image Lab software (Bio-Rad, Hercules, CA). For immunolocalization, leaf tissue was fixed and probed with primary antibodies against FLAG (1:40) and secondary goat anti-mouse Alexa Fluor 488-conjugated antibodies (ab150113, 1:200, Abcam) as described in *Ermakova et al., 2021*. Images were captured with a Zeiss 780 microscope using ZEN 2012 software (Black edition, Zeiss, Oberkochen, Germany). Images for plants of lines 27, 44, and azygous plants were acquired using online fingerprinting (488 nm excitation) with three user-defined spectral profiles for Alexa Fluor 488, endogenous autofluorescence, and chlorophyll. The spectral profile for endogenous autofluorescence was derived from the azygous control. The image for line 52 was initially collected as a full spectral scan (490–660 nm), then linearly un-mixed using the same online fingerprint settings as previously described. Images were post-processed with FIJI (*Schindelin et al., 2012*), and histograms for all images were min-max adjusted.

## Gas exchange measurements

Gas exchange and fluorescence analyses were performed at an irradiance of 1500 μmol m$^{-2}$ s$^{-1}$ (90% red/10% blue actinic light) and different intercellular $CO_2$ partial pressures using a LI-6800 (LI-COR Biosciences, Lincoln, NE) equipped with a fluorometer head 6800-01A (LI-COR Biosciences). Leaves were first equilibrated at 400 ppm $CO_2$ in the reference side, leaf temperature 25°C, 60% humidity, and flow rate of 500 μmol s$^{-1}$ and then a stepwise increase of $CO_2$ concentrations from 0 to 1600 ppm was imposed at 3-min intervals. Initial slopes of the $CO_2$ response curves were determined by linear fitting in OriginPro 2018b (OriginLab, Northampton, MA). Quantum yield of PSII upon the application of multiphase saturating pulses (8000 μmol m$^{-2}$ s$^{-1}$) was calculated according to *Genty et al., 1989*.

## C$^{18}$O$^{16}$O discrimination measurements

Simultaneous measurements of exchange of $CO_2$, $H_2O$, C$^{18}$O$^{16}$O, and H$_2$$^{18}$O were made by coupling two LI-6400XT gas exchange systems (LI-COR Biosciences) to a tunable diode laser (TDL; model TGA200A, Campbell Scientific Inc, Logan, UT) to measure C$^{18}$O$^{16}$O discrimination and a Cavity Ring-Down Spectrometer (L2130-i, Picarro Inc, Sunnyvale, CA) to measure the oxygen isotope composition of water vapor (*Osborn et al., 2017*). Measurements were made at 2% $O_2$, 380 μmol mol$^{-1}$ $CO_2$, leaf temperature of 25°C, irradiance of 1500 μmol m$^{-2}$ s$^{-1}$, and relative humidity of 55%. Each leaf was measured at 4 min intervals and 10 readings were taken. Mesophyll conductance was calculated as described by *Osborn et al., 2017* with the assumptions that there was sufficient CA in the mesophyll cytosol for isotopic equilibration between $CO_2$ and $HCO_3^-$. We also used calculations proposed by *Ogée et al., 2018* to estimate $g_m$. These calculations try to account for the rates of bicarbonate consumption by CA. We used the rate constant of CA hydration ($k_{CA}$) of 6.5 mol m$^{-2}$ s$^{-1}$ bar$^{-1}$ for these calculations.

## Statistical analysis

One-way and two-way ANOVAs with Tukey post hoc test were performed in OriginPro 2018b. A two-tailed, heteroscedastic Student's *t*-tests were performed in Microsoft Excel. Linear modeling was performed in R (*R Development Core Team, 2021*); mixed-effects models to test the need to incorporate transgenic event and copy status was performed in lme4 (*Bates et al., 2015*).

## Acknowledgements

The authors thank Xueqin Wang for *Setaria viridis* transformation, Zac Taylor for gas exchange measurements, Murray Badger and Dimitri Tolleter for measuring CA activity in yeast, Daryl Webb, Ayla Manwaring and the Centre for Advanced Microscopy at the Australian National University for confocal imaging, Wendy Sullivan for help with the stopped-flow spectrophotometry, and Nerea Ubierna for sharing the spreadsheet for the Ogee et al. $g_m$ calculations. This work is presented in the Australian provisional patent application # 2021900409.

## Additional information

### Funding

| Funder | Grant reference number | Author |
| --- | --- | --- |
| Australian Research Council | CE140100015 | Robert T Furbank |
| Australian Research Council | CE140100015 | Susanne von Caemmerer |
| Western Sydney University | VC Fellowship | Robert E Sharwood |
| Australian Research Council | DE130101760 | Robert E Sharwood |

The funders had no role in study design, data collection and interpretation, or the decision to submit the work for publication.

## Author contributions

Maria Ermakova, Conceptualization, Formal analysis, Investigation, Supervision, Visualization, Writing – original draft, Writing – review and editing, Data curation, Validation; Hannah Osborn, Formal analysis, Investigation, Visualization, Writing – original draft; Michael Groszmann, Formal analysis, Investigation, Supervision, Visualization, Writing – original draft, Writing – review and editing; Soumi Bala, Formal analysis, Investigation; Andrew Bowerman, Formal analysis, Methodology, Visualization, Validation, Writing – review and editing; Samantha McGaughey, Formal analysis, Investigation, Visualization, Writing – review and editing; Caitlin Byrt, Formal analysis, Supervision, Writing – review and editing; Hugo Alonso-cantabrana, Formal analysis, Supervision; Steve Tyerman, Methodology, Resources, Supervision, Writing – review and editing; Robert T Furbank, Conceptualization, Funding acquisition, Supervision, Writing – review and editing, Formal analysis; Robert E Sharwood, Conceptualization, Formal analysis, Funding acquisition, Investigation, Supervision, Visualization, Writing – original draft, Writing – review and editing; Susanne von Caemmerer, Conceptualization, Formal analysis, Funding acquisition, Investigation, Methodology, Supervision, Writing – original draft, Writing – review and editing, Project administration, Resources, Validation, Visualization

## Author ORCIDs

Maria Ermakova http://orcid.org/0000-0001-8466-4186
Michael Groszmann http://orcid.org/0000-0002-5015-6156
Andrew Bowerman http://orcid.org/0000-0003-1729-7843
Samantha McGaughey http://orcid.org/0000-0001-6133-0415
Caitlin Byrt http://orcid.org/0000-0001-8549-2873
Hugo Alonso-cantabrana http://orcid.org/0000-0002-5462-5861
Steve Tyerman http://orcid.org/0000-0003-2455-1643
Robert T Furbank http://orcid.org/0000-0001-8700-6613
Robert E Sharwood http://orcid.org/0000-0003-4993-3816
Susanne von Caemmerer http://orcid.org/0000-0002-8366-2071

## Decision letter and Author response

Decision letter https://doi.org/10.7554/eLife.70095.sa1
Author response https://doi.org/10.7554/eLife.70095.sa2

## Additional files

### Supplementary files

• Supplementary file 1. Amino acid similarity of *S. italica* and *S. viridis* PIPs.

• Supplementary file 2. Amino acid composition of *S. italica* PIPs at known substrate selectivity positions. Aquaporins have six transmembrane α-helices (H1-H6) joined by five loops (LA-LE). The monomeric channel is characterized by two NPA (Asn, Pro, Ala) motifs on loops A and E and along with the aromatic/arginine selectivity filter, these largely dictate the transport selectivity of substrates through the monomeric channels (*Azad et al., 2016*). Froger's positions indicate additional residues that are predicted for substrate specificity (*Froger et al., 1998*).

• Supplementary file 3. Primers used for qPCR.

• Transparent reporting form

### Data availability

All data generated or analysed during this study are included in the manuscript and supporting files.

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
