## [Editor Report]

The work generated and analysed transformants of Setaria with a putative CO_2_-permeable aquaporin and found evidence of improved photosynthesis and makes an contribution to understanding if aquaporins can be used to overcome limitations by mesophyll conductance. The new version is improved, making it clear that the variations are robust, but there are still some intriguing open biological questions.

---

## [Decision Letter]

**Decision letter after peer review:**

Thank you for submitting your article "Expression of a CO_2_-permeable aquaporin enhances mesophyll conductance in the C_4_ species *Setaria viridis*" for consideration by *eLife*. Your article has been reviewed by 3 peer reviewers, one of whom is a member of our Board of Reviewing Editors, and the evaluation has been overseen by Jürgen Kleine-Vehn as the Senior Editor. The following individual involved in review of your submission has agreed to reveal their identity: Berkley Walker (Reviewer #2).

All three reviewers found the question to be interesting. The work generated and analysed transformants of Setaria with a putative CO2-permeable aquaporin and found evidence of improved photosynthesis. The whole question of whether modulating aquaporins can overcome non-stomatal conductance is very exciting and important. All reviewers also felt that the text describes some nice physiological measurements on both yeast and plants, and these seem to support the idea.

However, all three reviews hit on major concerns, which could be addressed in the following ways:

Essential revisions:

1. There is a clear need to unambiguously establish the connection between the observed photosynthetic effects and the aquaporin content. This is especially important given that other efforts have run into the problem that the reported increases in photosynthesis could not be unambiguously attributed to the specific manipulations. Hopefully, the team will have saved back lines that could be tested relatively rapidly to address this issue.

2. The observed increase in photosynthesis at high Ci does not seem to fit into the expected model. If this cannot be explained, it suggests an unexpected, secondary effect. That would be also important, and might turn out to be more interesting in the long run, but would require some changes to the story.

There were also some interesting discussions covered in the specific reviews, e.g., the applicability of aquaporins to C3 versus C4 plants, which could add to the impact of the work.

*Reviewer #1 (Recommendations for the authors):*

The key issue that must be addressed is #2, above. It is possible that they have these kinds of data available already, but for some reason did not include it. If they do not, it would be very time consuming to produce it, or may not confirm the claims.

*Reviewer #2 (Recommendations for the authors):*

Can more be learned about the structure/function of the PIP proteins investigated as to why they had lower or higher capacity to transport CO2?

The proper controls are vital. Why was the Azygous plants of line 44 used? Was an empty vector or untransformed line also examined? Given the many instances of transgenic lines with initially-reported increases in photosynthesis that have not borne out in future studies, any additional controls would be helpful.

Is the quantification of the protein abundance of transgenes presented? This can be qualitatively seen from the westerns, but it would be more convincing if a clear relationship was shown between the increase in photosynthesis and the PIP protein abundance. This seems to be the case based on the description in the text, but if this was explicitly highlighted it would improve the argument made in the paper.

Line 135: Is it meant here that even though photosynthetic rates are not significantly higher (Line 131-133), there is a trend were the rates in the transgenic lines do appear to be higher?

More details are needed for the statistics used. For example, Figure 3 was analyzed with a one-way ANOVA with the genotype as the treatment. For this comparison, how were the different CO2 concentrations treated? Shouldn't a repeated two-way ANOVA be employed to look at the interactive effects of genotype and CO2 concentration while factoring in that each leaf was sampled at multiple CO2 concentrations? Also, the significant differences referred to in figure 4 could be more clearly indicated on the figure itself. The text clarifies things, but adding stars to the figures would help.

The initial portion of the discussion could effectively be employed in the introduction to introduce the reader unfamiliar with mesophyll conductance to its broad significance in photosynthesis and water relations.

*Reviewer #3 (Recommendations for the authors):*

This is an interesting and well-written paper. I have a number of points that the authors should consider.

1) In both Figure 1 (expression in yeast) and Figure 2 (expression in plants) although the transgene is present in the plasma membrane, it also seems to accumulate in other areas (spots in plant cells in Figure 2C?). Are these organelles? Is it possible that the ectopic expression is leading to altered CO2/water permeability in other membrane systems? It is of course difficult to prove a negative, but some comments/acknowledgement of this in the manuscript might be helpful.

2) Using the Flag-tag obviously helps visualise the transgene, but sometimes these tags{section sign} can interfere with protein function. Did the authors try making any transgenes without the flag-tag and, if so, were there more dramatic changes in phenotype?

3) I found line 13- 135 confusing.

Initially the authors state: "No significant changes were detected between transgenic and control plants in CO2 assimilation rates, effective quantum yield or stomatal conductance (Figure S7)", then in the next sentence "Since CO2 assimilation rate were consistently higher in all transgenic plants at low Ci (Figure 3)…"

I think they need to adjust lines 131-133- or have I missed something? I think line 135 is the important one- but can authors confirm/re-write.

4) As stated by the authors, the expectation is that in C4 plants increased mesophyll conductance will only improve assimilation rate at low Ci, yet my impression from the data in Figure 3 is that there is also a significant improvement at high Ci? Can the authors confirm this? If so, can they interpret? If the rest of the photosynthetic machinery is pretty much unchanged, how can this happen?

5) The data in Figure 4 a and b generally show quite a large spread within each line. This is typical for these type of experiments, particularly the gm analysis in part (b). This obviously means that the statistical underpinning is perhaps not the strongest, but given the difficulty of the experiments I though this was fine. I was more challenged by the data shown in Figure 4c where, due to the limited number of data points, it is possible to interpret the data several ways (e.g., line 52 suggests you can get a large shift in assimilation rate without major change in gm. With so few data points (and I acknowledge the amount of work in getting these data) I'm not sure this figure part is so convincing.

6) As noted on line 169, the relevance of the data to the physiological role of siPIP2:7 (and aquaprins in general) to mesophyll conductance is not really addressed by this paper. The authors should mention the recent (2020) paper by Kromdijk et al., in JExtBot indicating that in Arabidopsis a large number of aquaporins can be knocked out with minimal apparent outcome on mesophyll conductance. As the authors indicate, the PIPs used here is normally expressed in the root- so cannot play a normal role in CO2 flux (it is an ectopic expression of the gene). Acknowledgement of some of the continuing controversy on the endogenous importance of these transporters in mesophyll conductane would be wise.

---

## [Author Response]

Essential revisions:1. There is a clear need to unambiguously establish the connection between the observed photosynthetic effects and the aquaporin content. This is especially important given that other efforts have run into the problem that the reported increases in photosynthesis could not be unambiguously attributed to the specific manipulations. Hopefully, the team will have saved back lines that could be tested relatively rapidly to address this issue.2. The observed increase in photosynthesis at high Ci does not seem to fit into the expected model. If this cannot be explained, it suggests an unexpected, secondary effect. That would be also important, and might turn out to be more interesting in the long run, but would require some changes to the story.

We thank the reviewers for these suggestions and we agree these are important issues to address. We have now added Figure 3—figure supplement 2 to the manuscript where we explored the relationship between the relative abundance of PIP2;7 and the initial slopes/saturating parts of the ACi curves in individual plants of four transgenic lines. Modelling of the initial slope of the ACi curves compared to sample-specific protein accumulation found a significant second order polynomial relationship (R^2^ = 0.345, p < 0.05, F = 5.007, df = 2 and 19) which was significantly better than that achieved using a linear model (p < 0.05). No improvement to these models could be achieved by addition of random effects accounting for transgenic event or zygosity of samples.

Moreover, Figure 3—figure supplement 2c unambiguously demonstrates that PIP2;7 expression does not influence the maximum rates of assimilation (*A*_max_), as no significant relationship could be found using linear and mixed effects modelling methods. Dr Andrew Bowerman has performed this statistical analysis in R and joined as a co-author.

Interestingly, the highest expressing plants did not show a further increase of the initial slope, as a predicted maximum effect was found to be at 1.65 (r.u.) protein content. This suggests a presence of confounding factor(s) which might include but not limited to:

1) Transport of unknown substrate(s) by PIP2;7;

2) Ectopic expression of aquaporins in other tissues affecting water relationship of leaves;

3) Too strong expression preventing the correct targeting/function of other membrane proteins;

4) Different efficiency of PIP2;7 targeting to the plasma membrane between the lines/plants/leaves/cells;

5) Unknown ‘baseline’ of aquaporin-mediated CO_2_-permeability in Setaria leaves.

This discussion has now been added to P 8 LL 229-236.

Unlike other published research reporting increases in photosynthesis that sometimes could not be attributed to the specific manipulations, our conclusions are supported by functional and physiological analyses, cellular localization studies and mathematical modelling. We provide evidence that:

1) PIP2;7 is a functional CO_2_ pore in heterologous expression system;

2) In transgenic plants, PIP2;7 localizes to the mesophyll plasma membranes where it’s CO_2_ permeability could potentially contribute to the mesophyll conductance;

3) In line with the biochemical model of C_4_ photosynthesis, PIP2;7-mediated increase of mesophyll conductance affects the initial slopes of the ACi curves but not the saturating rates of assimilation.

Taken together, our results confirm that PIP2;7 is a functional CO_2_ pore in vivo and show the effect of increased mesophyll conductance on C_4_ photosynthesis.

There were also some interesting discussions covered in the specific reviews, e.g., the applicability of aquaporins to C3 versus C4 plants, which could add to the impact of the work.

Indeed, works investigating the significance of CO_2_ permeable aquaporins in C_3_ plants have sometimes reached conflicting conclusions, emphasizing the complexity of plant aquaporin research. These discrepancies are likely due to redundancy of multiple aquaporin isoforms and confounding factors discussed above. We have now expanded this discussion (P 7 LL 183-196).

Reviewer #1 (Recommendations for the authors):The key issue that must be addressed is #2, above. It is possible that they have these kinds of data available already, but for some reason did not include it. If they do not, it would be very time consuming to produce it, or may not confirm the claims.

We provided the details on lines selection in the result section (P 5 LL 131-132): “lines 27, 44 and 52 were selected for further analysis because they had the strongest FLAG signal per transgene insertion number” and this information has now also been added to the Materials and Methods (P 11 LL 309-312). This, as Reviewer 2 mentioned, is one of the strengths of our work since we selected the lines based on the expression level and not a photosynthetic phenotype. To further support our conclusions, we have now added Figure 3—figure supplement 2 demonstrating protein and gas-exchange analysis of T_1_ plants from four independent lines (with additional line 49, azygous plants of line 49 and WT).

Results presented on Figure 3—figure supplement 2 suggest that including in the analysis additional lines with relatively lower expression levels would likely not be helpful since the strongest increase of the initial slope was achieved in plants with medium-high level of expression.

Reviewer #2 (Recommendations for the authors):Can more be learned about the structure/function of the PIP proteins investigated as to why they had lower or higher capacity to transport CO2?

Predicting selectivity of aquaporins based on their sequence/structure is, indeed, an exciting possibility. Based on previous functional studies, there are now consensus sequence signatures associated with different substrates. In Supplementary File 2 we show that eight SiPIPs, including PIP2;7, match the current consensus sequence for CO_2_ transport. Thus, although our work supports the current CO_2_ sequence, more large-scale functional studies are required to narrow down the consensus signatures for multiples substrates.

The proper controls are vital. Why was the Azygous plants of line 44 used? Was an empty vector or untranformed line also examined? Given the many instances of transgenic lines with initially-reported increases in photosynthesis that have not borne out in future studies, any additional controls would be helpful.

We agree with the reviewer. Choosing the right controls is complicated and largely depends on the model species and a nature of experiments. Based on our previous experience with *Setaria viridis* transgenics, azygous lines that went through the process of transformation but do not exhibit positional effects of insertions that are possible in the empty vector lines, present a good compromise. WT and azygous plants of other lines, included in some experiments as additional controls, typically exhibited phenotypes of control plants (azygous line 44) and thus were not reported in this work.

Is the quantification of the protein abundance of transgenes presented? This can be qualitatively seen from the westerns, but it would be more convincing if a clear relationship was shown between the increase in photosynthesis and the PIP protein abundance. This seems to be the case based on the description in the text, but if this was explicitly highlighted it would improve the argument made in the paper.

We now show the detailed analysis of the PIP2;7 abundance and the initial slopes of the ACi curves on Figure 3—figure supplement 2.

Line 135: Is it meant here that even though photosynthetic rates are not significantly higher (Line 131-133), there is a trend were the rates in the transgenic lines do appear to be higher?

We have now amended that sentence to avoid confusions (P 6 L 159).

More details are needed for the statistics used. For example, Figure 3 was analyzed with a one-way ANOVA with the genotype as the treatment. For this comparison, how were the different CO2 concentrations treated? Shouldn't a repeated two-way ANOVA be employed to look at the interactive effects of genotype and CO2 concentration while factoring in that each leaf was sampled at multiple CO2 concentrations? Also, the significant differences referred to in figure 4 could be more clearly indicated on the figure itself. The text clarifies things, but adding stars to the figures would help.

Following recommendations of the reviewer, we performed the two-way ANOVA on the ACi curves but have not found significant differences. Significant differences between the lines reported on Figure 4 are now indicated with asterisks.

The initial portion of the discussion could effectively be employed in the introduction to introduce the reader unfamiliar with mesophyll conductance to its broad significance in photosynthesis and water relations.

We have now added this information to the introduction (P 3 LL 51-56).

Reviewer #3 (Recommendations for the authors):This is an interesting and well-written paper. I have a number of points that the authors should consider.1) In both Figure 1 (expression in yeast) and Figure 2 (expression in plants) although the transgene is present in the plasma membrane, it also seems to accumulate in other areas (spots in plant cells in Figure 2C?). Are these organelles? Is it possible that the ectopic expression is leading to altered CO2/water permeability in other membrane systems? It is of course difficult to prove a negative, but some comments/acknowledgement of this in the manuscript might be helpful.

We agree with the reviewer, unavoidable ectopic expression of PIP2;7 in other cells/compartments combined with the potential to transport yet unknown substrates present a major confounding factor complicating the assessment of transgenic plants. This effect is likely demonstrated by the relationship between the PIP2;7 abundance and the initial slopes of the ACi curves where the highest expressing plants lack the phenotype (Figure 3—figure supplement 2; as discussed in essential revisions). This discussion has now been added to P 8 LL 229-236.

2) Using the Flag-tag obviously helps visualise the transgene, but sometimes these tags{section sign} can interfere with protein function. Did the authors try making any transgenes without the flag-tag and, if so, were there more dramatic changes in phenotype?

We did not attempt to overexpress aquaporin without the FLAG tag since, as reviewer mentioned, it would make impossible verifying the targeting of PIP2;7 to the mesophyll plasma membrane. However, it does present an interesting opportunity and, with the developing of specific antibodies against PIP2;7, this experiment can be performed in future.

3) I found line 13- 135 confusing.Initially the authors state: "No significant changes were detected between transgenic and control plants in CO2 assimilation rates, effective quantum yield or stomatal conductance (Figure S7)", then in the next sentence "Since CO2 assimilation rate were consistently higherin all transgenic plants at low Ci (Figure 3)…"I think they need to adjust lines 131-133- or have I missed something? I think line 135 is the important one- but can authors confirm/re-write.

We have now amended that sentence to avoid confusions (P 6 L 159).

4) As stated by the authors, the expectation is that in C4 plants increased mesophyll conductance will only improve assimilation rate at low Ci, yet my impression from the data in Figure 3 is that there is also a significant improvement at high Ci? Can the authors confirm this? If so, can they interpret? If the rest of the photosynthetic machinery is pretty much unchanged, how can this happen?

This issue has been addressed in the essential revisions.

5) The data in Figure 4 a and b generally show quite a large spread within each line. This is typical for these type of experiments, particularly the gm analysis in part (b). This obviously means that the statistical underpinning is perhaps not the strongest, but given the difficulty of the experiments I though this was fine. I was more challenged by the data shown in Figure 4c where, due to the limited number of data points, it is possible to interpret the data several ways (e.g., line 52 suggests you can get a large shift in assimilation rate without major change in gm. With so few data points (and I acknowledge the amount of work in getting these data) I'm not sure this figure part is so convincing.

We appreciate that reviewer understands the difficulties of measuring mesophyll conductance in C_4_ plants. The spread of data and differences between the lines, mentioned by the reviewer, are likely caused by the confounding factors of PIP2;7 expression already discussed above. We have now expanded the discussion to clarify these results (P 8 LL 229-236) Although it doesn’t present executive evidence, we think that relating our results to the mathematical model of C_4_ photosynthesis does provide an additional link in supporting our conclusions (as discussed in the essential revisions). It is also important for mathematical models to be validated by experimental results. Our results generally support the von Caemmerer and Furbank model (Susanne von Caemmerer, 2000; S. von Caemmerer and Furbank, 1999) once again showing the importance of this classical model for understanding C_4_ photosynthesis.

6) As noted on line 169, the relevance of the data to the physiological role of siPIP2:7 (and aquaprins in general) to mesophyll conductance is not really addressed by this paper. The authors should mention the recent (2020) paper by Kromdijk et al., in JExtBot indicating that in Arabidopsis a large number of aquaporins can be knocked out with minimal apparent outcome on mesophyll conductance. As the authors indicate, the PIPs used here is normally expressed in the root- so cannot play a normal role in CO2 flux (it is an ectopic expression of the gene). Acknowledgement of some of the continuing controversy on the endogenous importance of these transporters in mesophyll conductane would be wise.

We agree with the reviewer and have now expanded this discussion (P 7 LL 183-196).